# The Precision, Inter-Rater Reliability, and Accuracy of a Handheld Scanner Equipped with a Light Detection and Ranging Sensor in Measuring Parts of the Body—A Preliminary Validation Study

**DOI:** 10.3390/s24020500

**Published:** 2024-01-13

**Authors:** Enrica Callegari, Jacopo Agnolucci, Francesco Angiola, Paolo Fais, Arianna Giorgetti, Chiara Giraudo, Guido Viel, Giovanni Cecchetto

**Affiliations:** 1Unit of Legal Medicine and Toxicology, Department of Cardiac, Thoracic, Vascular Sciences and Public Health, University of Padova, Via Falloppio 50, 35100 Padova, Italy; enricacallegari.ec@gmail.com (E.C.); jacopo.agnolucci@gmail.com (J.A.); angiolafrancesco6@gmail.com (F.A.); guido.viel@unipd.it (G.V.); 2Unit of Legal Medicine, Department of Medical and Surgical Sciences, University of Bologna, Via Zamboni 33, 40126 Bologna, Italy; paolo.fais@unibo.it (P.F.); ari.giorgetti@gmail.com (A.G.); 3Unit of Radiology, Department of Cardiac, Thoracic, Vascular Sciences and Public Health, University of Padova, Via Giustiniani 2, 35100 Padova, Italy; chiara.giraudo@unipd.it

**Keywords:** handheld scanner, LiDAR, anthropometric measurements, 3D model reconstruction

## Abstract

Background: Anthropometric measurements play a crucial role in medico-legal practices. Actually, several scanning technologies are employed in post-mortem investigations for forensic anthropological measurements. This study aims to evaluate the precision, inter-rater reliability, and accuracy of a handheld scanner in measuring various body parts. Methods: Three independent raters measured seven longitudinal distances using an iPad Pro equipped with a LiDAR sensor and specific software. These measurements were statistically compared to manual measurements conducted by an operator using a laser level and a meterstick (considered the gold standard). Results: The Friedman test revealed minimal intra-rater variability in digital measurements. Inter-rater variability analysis yielded an ICC = 1, signifying high agreement among the three independent raters. Additionally, the accuracy of digital measurements displayed errors below 1.5%. Conclusions: Preliminary findings demonstrate that the pairing of LiDAR technology with the Polycam app (ver. 3.2.11) and subsequent digital measurements with the MeshLab software (ver. 2022.02) exhibits high precision, inter-rater agreement, and accuracy. Handheld scanners show potential in forensic anthropology due to their simplicity, affordability, and portability. However, further validation studies under real-world conditions are essential to establish the reliability and effectiveness of handheld scanners in medico-legal settings.

## 1. Introduction

Anthropometric measurements play a crucial role in medico-legal activities commonly conducted during forensic investigations. They serve multiple purposes, including identifying individuals or corpses [1,2,3,4], aiding in cranio-facial reconstructions [5], and providing essential data for the reconstruction of shooting crimes [6,7,8], road accidents [9,10], precipitations [11], or any other traumatic injuries [12]. It is strongly recommended to conduct these measurements during forensic autopsies [13,14] and, if feasible, during death scene investigations [15]. In particular, anthropometric measurements are usually taken from the heel in order to determine the “height” of the subject. Traffic accidents, in which it is necessary to determine the dynamics of the accident, and falls from a height, where it is necessary to calculate the center of gravity, are examples of situations where longitudinal anthropometric measurements are essential. As a result, the precision and accuracy of these measurements are of utmost importance in forensic medicine due to their potential impact at trial.

To date, anthropometric measurements have been carried out manually using physical metersticks or measuring tapes [16]. In accordance with international medico-legal recommendations [1,13,14], the hell–head measurement should always be performed in a forensic post-mortem examination. It is important to note, however, that in certain circumstances, such as in the case of unidentified corpses, many other anthropometric measurements must be performed [17]. In addition to anthropometric measurements, international guidelines recommend that injuries be classified topographically by measuring the distance between injuries and anatomical landmarks. However, it may be difficult to take all these measurements with physical metersticks or measuring tapes in some instances, such as during a medico-legal death scene investigation (DSI), since it would take a significant amount of time. In addition, several aspects that might necessitate anthropometric measurements often become apparent only after the corpse has been examined or autopsied.

More recently, digital devices such as three-dimensional computer-aided design (3D/CAD)-supported photogrammetry [18] and structured-light 3D scanners [19,20] have been utilized for the identification and determination of anthropometric parameters through digital measurements. In the clinical setting, for the purpose of monitoring physical activity and dietary interventions, a commercial three-dimensional optical (3DO) imaging system has been evaluated for assessing total and regional body composition and anthropometric measurements, providing precise and accurate estimates [21]. Structured-light 3D scanners are relatively new technologies that are mostly employed in clinical settings [22]. A major difference between scanning protocols for living people and the present study is that since this method involves acquiring scans of a cadaver, any motion artifacts that may exist in the case of living people are inherently eliminated. Computed tomography (CT) is commonly employed in post-mortem investigations [23,24,25,26], and its use for forensic anthropological measurements has recently been tested [27,28]. Fourie et al. evaluated the accuracy and reliability of standard anthropometric linear measurements on cadaver heads made by three different three-dimensional scanning systems (surface laser scanning (Minolta Vivid 900), cone beam computed tomography (CBCT), and 3D stereophotogrammetry (Di3D system)) and compared them with physical linear measurements, showing that measurements recorded by the three 3D systems appeared to be both sufficiently accurate and reliable enough for research and clinical use [18]. It is notable that there are plenty of technologies that can produce 3D scans of objects and bodies, many of which are commonly employed during forensic autopsies, above all in Switzerland [20]. However, the above-mentioned technologies are expensive, need specific personnel training, and are of large dimensions (not portable and therefore not usable at DSI). 

In the last few years, smaller, portable handheld scanners, such as smartphones and tablets, capable of performing 3D scans of objects and spaces have been developed and commercialized. These new devices have the advantages of being portable, low-cost, and easily usable by anyone. The LiDAR technology [29] employs a laser system that can calculate the distance from objects by calculating the time running from the emission of the laser impulse and the reception of the same signal going back to the sensor of the device. This technology is widely employed in seismology, archaeology, geology, architecture, and meteorology.

Recently, Maiese et al. [30] used an iPad Pro equipped with a LiDAR sensor to elaborate 3D models obtained during ten autopsies in order to evaluate the quality and trustworthiness of these models compared to conventional autopsy photographic records. According to Fineschi et al., two apps were used to acquire scans (TRNIO and 3D Scanner), and the 3D models acquired were then processed using the MeshLab software. The above study concluded that this new handheld scanner could serve as a valuable tool in the forensic field for documenting autopsy findings and capturing high-quality 3D images. Thus, a single scan acquisition of the entire three-dimensional surface of the cadaver could allow all anthropometric measurements to be taken digitally at a time following the medico-legal DSI or autopsy of the cadaver, as well as be able to be repeatable whenever necessary. Nonetheless, currently, there is a lack of studies testing the reliability of such devices in comparison with the current medico-legal “gold standard” (i.e., manual measurements).

Therefore, the current study aimed to assess the precision, inter-rater reliability, and accuracy of an iPad Pro equipped with a LiDAR sensor and paired with the Polycam app in conjunction with the MeshLab software for measuring several parts of the body.

## 2. Materials and Methods

### 2.1. Study Design

This study was conducted in accordance with local ethical rules, and permission was obtained from the public prosecutor to perform measurements on a body undergoing forensic autopsy at an institute of legal medicine in the northeastern region of Italy.

A highly experienced forensic pathologist (GC) with 20 years of expertise conducted ten sets of manual anthropometric measurements. These measurements comprised 7 longitudinal distances (Figure 1) using a laser level and a meterstick. The purpose of considering these specific 7 longitudinal distances was to evaluate the precision and accuracy of measurements for both short and long distances, as is usually employed in forensic anthropology. The same operator utilized the iPad Pro (details provided below) to scan the body ten times, with an average scanning time of approximately 20 s per scan. For each body scan and anthropometric measure tested, three independent blind raters conducted digital measurements ten times using three different computers with the same software installed (see details below). These repetitions were performed on different days, with a minimum interval of 24 h between each session. The blind raters consisted of a forensic pathologist with 15 years of experience (GV), a forensic pathologist specializing in anthropology (PF), and a resident in forensic medicine (EC). EC was specifically trained by the senior forensic pathologist (GC) in utilizing the MeshLab software—an open-source tool developed by the Institute of Information Science and Technologies, Consiglio Nazionale delle Ricerche, Rome, Italy. Each rater performed 700 digital measurements (seven distances tested on ten different scans, each measured ten times), resulting in a total of 2100 measurements.

Subsequently, the following parameters were calculated and assessed [31]:The precision or intra-rater variability, defined as the degree to which repeated measures performed in different moments produce similar results, was tested by comparing, for each anthropometric measure, the measures performed across the ten days;The inter-rater reliability, defined as the degree of concordance among different raters who analyze the same parameter independently of each other, was tested by comparing, for each anthropometric measure, the measures performed across the ten days by the 3 involved raters and by comparing the median values obtained by each rater for all the anthropometric measures using the intraclass correlation coefficient (ICC);Accuracy of the handheld scanner, indicating the closeness between the true value and the value measured with the investigated technique, was evaluated by determining the percentage error between the real (manual) measurement and the median of each digital distance measured by the three raters involved (resulting in 100 measurements for each anthropometric measure).

### 2.2. Manual Measurements

The senior pathologist (GC) conducted the following steps with the body positioned supine on the evisceration table in the autopsy room:Identified and marked 6 anthropometric landmarks on the body using an “X” (Figure 1);Measured 7 longitudinal distances using a 2-meter rigid meter placed beside the corpse, aligned parallel to its main axis. Additionally, a laser level (Laser Level mode CM-701, Cigman, Essen, Germany) was used to facilitate the orthogonal transposition of the specifically marked landmarks. Each longitudinal distance was measured ten times.

### 2.3. Body Scan

An iPad Pro (2020—Apple Inc., Cupertino, CA, USA), an 11-inch tablet operating on the iPadOS mobile system, was utilized in this study. It was equipped with a LiDAR (“Light Detection and Ranging”) sensor, a pulsed laser that records the time taken (at nanosecond speeds) for the signal to return to its source, allowing the generation of 3D reconstructions. The chosen software was Polycam (ver. 3.2.11, Polycam Inc., Altadena, CA, USA), a paid application designed to create high-quality 3D models from scans of spaces or objects using the LiDAR sensor.

The ten scans were conducted by a forensic pathologist with 20 years of experience (GC). GC held the device approximately 1.5 m above the body. By moving the device along the longitudinal axis of the corpse, GC performed a scan of the corpse from head to toe along its midline. Minor lateral movements were made to ensure better scanning of the body’s lateral surfaces. Each scan was processed using the “object” processing option and saved as a .obj file. This procedure takes only a few minutes (about 1–3 min).

### 2.4. Digital Measurements

The data stored on the handheld scanner were transferred to a PC through wireless transfer options. Subsequently, utilizing the MeshLab software (ver. 2022.02), the three aforementioned blind raters conducted measurements for each of the seven selected distances ten times over ten different days (one measurement per day).

By selecting “draw XYZ axes in world coordinates”, a Cartesian system became visible on the scan (see Figure 2a,b). The “manipulator” tool was used to position the scanned body parallel to the X-axis and perpendicular to the Z-axis (the Z-axis was tangential to the toes).

The “reference scene tool” allowed for the selection of two points, the coordinates of which were identified on the X-axis and Z-axis (Figure 2c,d), in order to calculate the distance between the two landmarks. The measurements were conducted by transposing the coordinates of the two landmarks specifically along the X-axis.

### 2.5. Statistical Analyses

The data were collected in an Excel^®^ database (Microsoft Corporation^®^, version 2309 Build 16.0.16827.20166) and analyzed using SPSS Statistics^®^ (IBM Corp, version 26.0.0.0). Graphical representations of the results of the statistical analyses were performed using Prism (GraphPad Software, LLC, version 10.0.0.3).

Non-parametric statistics were applied since a parametric distribution for each set of data could not be proved by scatterplot. Anthropometric measurements obtained with the laser level over the 10-day period were described by median and interquartile range. To compare these measurements, a non-parametric one-way ANOVA for matched or repeated measures (Friedman test) was utilized.

For assessing precision or intra-rater variability, both analog measurements acquired during the autopsy and digital measurements obtained by each rater were compared. The non-parametric one-way ANOVA for matched or repeated measures (Friedman test) was applied, considering a statistically significant difference with a *p*-value < 0.05.

Median values of measurements obtained by each operator were utilized for subsequent statistical analyses. Friedman test with multiple comparisons and the intraclass correlation coefficient (ICC) were used to evaluate the inter-rater reliability on the median distances measured by the 3 raters, with the Friedman test being performed separately for each anthropometric measure, while the ICC is performed on the whole set. For the ICC, a two-way mixed model was set, which accounts for “fixed” raters and variable measurements, while also checking for absolute agreement. Based on the 95% confidence interval of the ICC estimate, values less than 0.500, between 0.500 and 0.750, between 0.750 and 0.900, and greater than 0.900 were considered indicative of poor, moderate, good, and excellent reliability, respectively [32].

Finally, data from digital measurements were compared to the gold standard value (median of manual measurements) to calculate the percentage of error and evaluate accuracy. Accuracy was assessed for each rater in terms of maximum and median percent error across 10 measurements. Additionally, the median accuracy among the three raters was computed (Equation (1) below), considering percent errors below 2% as acceptable and below 1.5% as excellent [33].
(1)$$\mathrm{Median}\;\mathrm{accuracy}=\;\frac{\mathrm{Median}\;\mathrm{of}\;\mathrm{digital}\;\mathrm{measurements}\;\mathrm{of}\;\mathrm{the}\;3\;\mathrm{raters}\;-\;\mathrm{Median}\;\mathrm{of}\;\mathrm{manual}\;\mathrm{measurements}\;}{\mathrm{Median}\;\mathrm{of}\;\mathrm{manual}\;\mathrm{measurements}}\;\times \;100\;$$

## 3. Results

The results are reported in Table 1 and Table 2.

Measurements obtained with the laser level exhibited negligible intra-rater variability (*p* = 0.0538).

Concerning digital measurements, the Friedman test did not find any statistically significant difference among repeated measures performed by the same operator (intra-rater variability), except for the anthropometric measure E (distance between heel and chin) for operator 3 (*p* = 0.0136) (Table 1). 

Assessment of inter-rater variability via Friedman test with multiple comparisons highlighted statistically significant differences in anthropometric measure A (heel–knee, Rater 1 vs. Rater 2; *p* = 0.0110), anthropometric measure E (heel–chin, Rater 1 vs. Rater 3; *p* = 0.0001), and anthropometric measure F (heel–nasion, Rater 2 vs. Rater 3; *p* = 0.0010). A graphical representation is shown in Figure 3. For the other anthropometric measures, inter-rater differences were consistently non-significant (*p* > 0.05). The level of agreement among the measurements, expressed with the ICC, was 1.00, demonstrating an excellent correlation among the measurements across all operators (Table 1).

Regarding accuracy, the median percentage error remained under 1% for all measurements (Table 2, Figure 4). The highest percent error value was observed for Rater 3 in the determination of the anthropometric measure A (about 1.2%), while the lowest value was obtained by Rater 1 in the determination of the anthropometric measures F and G (about 0.2% in both cases). Rater 2 and Rater 3 exhibited slightly lower accuracies compared to Rater 1. Among the maximum median percentage errors across the different raters, anthropometric measures A and B showed the highest errors (about 0.6% and 0.4%, respectively), while the lowest value was obtained for anthropometric measures E, F, and G (about 0.1%). 

In forensic anthropology, errors are generally considered relative and not absolute. Additionally, the acceptable range for the relative, or percent, error in anthropometry is <2% [33]. Our results show a relative error always below 1.5%.

## 4. Discussion

In the forensic field, anthropometric measures of living or deceased individuals may play an essential role in multiple situations. To date, anthropometric measurements have been carried out manually using physical metersticks or measuring tapes [16]. However, more recently, digital devices such as three-dimensional computer-aided design (3D/CAD)-supported photogrammetry [18] and structured-light 3D scanners [19,20] have been utilized, albeit less frequently due to their lack of portability and high cost in terms of both equipment and training required for personnel.

Over the past few years, the availability of measuring instruments has significantly expanded. Specifically, numerous portable scanners and image acquisition tools, termed handheld scanners, have been developed for widely used devices like smartphones and tablets. These new devices have the advantages of being portable, low-cost, and easily used by anyone. A recent study [30] tested an Apple^®^ tablet equipped with a LiDAR sensor, generating three-dimensional (3D) reconstructions of ten bodies directly within the autopsy room. The study concluded that this new handheld scanner could serve as a valuable tool in the forensic field for documenting autopsy findings and capturing high-quality 3D images.

In this study, the employment of a laser level allowed for the orthogonal transposition of the body landmarks on the rigid meterstick, allowing the measurements to be detected on the same longitudinal axis (Figure 1). This was necessary because as much as the use of metersticks and calipers represents the gold standard for anthropometric measurements, manual measurements conducted between two body landmarks can lack precision, as these landmarks might be situated at different levels of the body. Thus, these instruments should always be used in conjunction with other devices (e.g., the laser level); however, this is not always possible, especially in certain circumstances (e.g., during a medico-legal death scene investigation—DSI). As a result of the present study, the measurement performed by matching the meterstick and the laser level exhibited statistically insignificant intra-rater variability. This confirmed that the manual measurement method was precise and, hence, that it could be used to evaluate the precision of the digital measurements.

In our study, we tested the same type of tablet used by Maiese et al. [30], paired with a different app (the Polycam app) and the same MeshLab software, in order to identify accuracy and precision in anthropometric digital measurements compared to the manual measurements performed with a laser level and meterstick.

For digital measurements, the Friedman test revealed negligible intra-rater variability in digital measurements, except for one instance (E) with operator 3 (a resident in forensic medicine). However, among the measurements conducted by the senior pathologist, no significant differences were observed. These results seem to suggest a lower intra-rater variability for trained operators.

Concerning the inter-rater variability, some anthropometric measures, namely A (heel–knee), E (heel–chin), and F (heel–nasion), exhibited greater inter-rater variability than others. Since the anthropometric landmarks had been previously marked using an “X”, it is unlikely that the disagreements introduced into the measurement process are due to the lack of recognized reference points on the body. Notably, experience in forensic pathology seemed to influence precision, as experienced raters exhibited fewer discrepancies. Despite slight differences among raters, overall agreement remained excellent (ICC = 1), indicating high concordance among the three independent blind raters.

Our results regarding the variability of intra- and inter-rater digital measurements obtained by the proposed method indicate that experience in forensic pathology implies more precise digital measurements. This method, however, allows even less experienced assessors to obtain sufficiently precise results. In light of the above, this method can be used to determine digital anthropometric measurements by personnel with various levels of forensic expertise, with the understanding that forensic expertise is associated with more precise measurements.

Finally, the accuracy of digital measurements showed errors below 1.5%, with smaller anthropometric measures displaying slightly higher inaccuracies. The greatest inaccuracies were observed in the shortest anthropometric measures (A and B), stemming from errors primarily within a range of a few millimeters. Errors were calculated by comparing the median of digital measurements with the median of manual measurements. We observed that smaller distances might be affected by more relevant errors. According to the forensic anthropological literature and in line with the results obtained for the accuracy of digital measurements, any error of less than 1.5% is not considered potentially significant. Further experiments should be devoted to measuring the percentage error for very short anthropometric distances or small injuries. Indeed, the method examined here might be less suitable for documenting small injuries or short anthropometric measures. In addition, the results indicate that the error rate, although always <1.5%, differed among the three blind raters. This may be the result of their different experience and expertise in forensic pathology. Furthermore, it is important to note that anthropometric measurements are subject to an ineradicable error. Also, when other methods of measurement (e.g., metersticks, post-mortem CT, 3D stereophotogrammetry [18]) have been employed, intra-operator and inter-operator errors have always been registered.

This preliminary study concluded that the LiDAR technology paired with the Polycam app and subsequent digital measurements with the MeshLab software demonstrate high precision, inter-rater agreement, and accuracy, showcasing potential in forensic applications. Scans that were acquired using such types of technology and the Polycam app generated three-dimensional models comprising polygonal meshes, with 3D reconstructions textured faithfully. This fidelity was due to the seamless integration of the camera system, applying photographic textures onto the mesh. Additionally, while conducting scans with the LiDAR system, the senior pathologist (GC) discovered its capability to move across various planes and rotate or oscillate, facilitating the scanning of lateral surfaces of the body without disrupting the scan or texture.

The use of this technology for anthropometric measurement has several advantages. In particular, during the medico-legal DSI or autopsy, only scanning the cadaver will be required (an easy and quick process) since digital anthropometric measurements might be obtained at a later stage; conversely, with the physical meterstick, all measurements must be taken immediately, which is time-consuming and not always possible, especially in an urgent situation as a DSI. Additionally, digital scans can be retained for future measurements that may be required, shared with consultants to take measurements, even after years or when far away from the scene, presented to jurors to support expert witnesses [34], as well as used for educational purposes.

The current study also has certain limitations. First of all, it relies on the senior pathologist who conducted the 10 body scans and marked the anthropometric landmarks before the raters performed digital measurements. In a real forensic context, if individual raters were tasked with scanning the body and selecting anthropometric landmarks, we might expect greater inter-rater variability and reduced accuracy. Under these circumstances, we anticipate that the raters’ experience in forensic pathology would significantly influence the outcomes in terms of accuracy and precision. To date, there are no fully automated methods for taking anthropological measurements. In the future, it would be interesting to develop forensic-specific hardware and software capable of acquiring automated anthropometric measurements, thereby reducing the need for a human intervention. These technologies, however, are not commercially available at the present time. Currently, these devices utilize software such as Meshlab, designed for architects and designers. However, they require a basic understanding of math and geometry and are not optimized for measuring the body and its anatomical parts. As previously indicated [30], optimizing the interface between hardware and software is crucial to achieving optimal performance with the LiDAR sensor. For this reason, it will be necessary to develop a software specifically dedicated to forensic anthropology and pathology. Secondly, our study solely considered relatively long distances (always > 40 cm), showing that smaller anthropometric measurements are subject to a higher percentage of error. Studying the use of this technology to determine smaller distances in depth would be of interest, as this technology may be used in forensic pediatrics to characterize traumatic injuries and their location with respect to landmarks. Additionally, the technology was tested under standard conditions (e.g., in the sector room, with a clean cadaver that showed no signs of putrefaction or major injuries). However, the “real life” conditions (e.g., light conditions, dirt skin, absence of landmarks, etc.) should negatively influence the precision and accuracy of this method. Lastly, the raters involved had no prior training with the Polycam and MeshLab softwares. However, it is expected that their performance would improve with familiarity and experience gained over a learning period.

More studies on a real forensic setting, especially investigating the impact of a training period and of the forensic experience of different raters (with and without a period of specific training), are needed in order to assess the true reliability of the proposed method based on a handheld scanner equipped with a LiDAR sensor.

In conclusion, handheld scanners offer promise in forensic anthropology due to their simplicity, affordability, and portability. Developing dedicated scan applications and measurement software tailored for medico-legal purposes (i.e., optimized for small curved objects or small body parts, such as fingers, hands, nose, and ears) could significantly enhance their effectiveness in forensic investigations.

Further validation studies under real-world conditions are necessary to fully establish the reliability and efficiency of handheld scanners in forensic settings. Additionally, optimizing hardware–software interfaces and developing dedicated apps for anatomical measurements are essential steps toward improving their forensic utility.

## Figures and Tables

**Figure 1 sensors-24-00500-f001:**
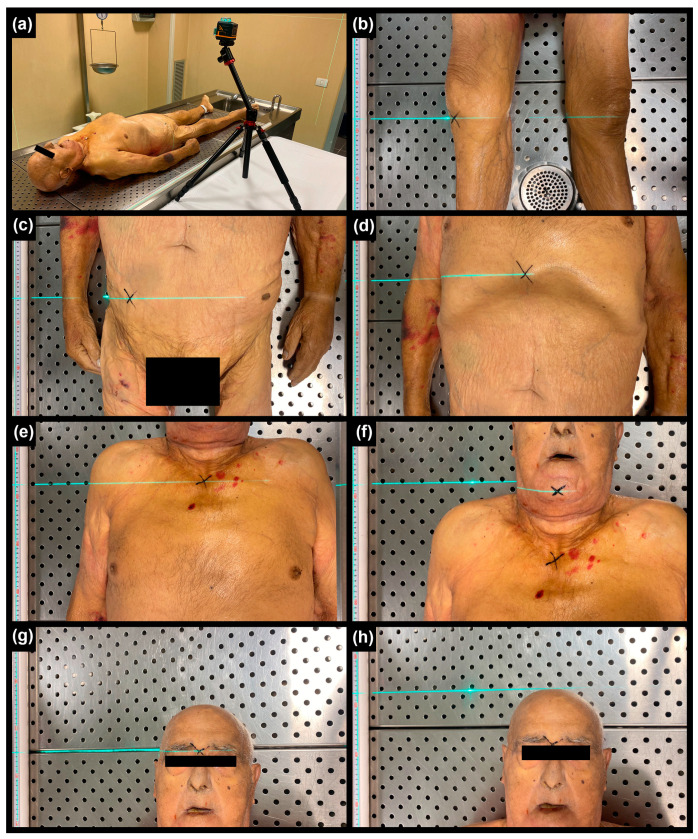
Rigid meter placed beside the corpse and laser level used to facilitate the orthogonal transposition of the marked landmarks (**a**); heel–knee (**b**); heel–superior iliac spine (**c**); heel–xiphoid process (**d**); heel–jugule (**e**); heel–chin (**f**); heel–nasion (**g**); heel–head (**h**).

**Figure 2 sensors-24-00500-f002:**
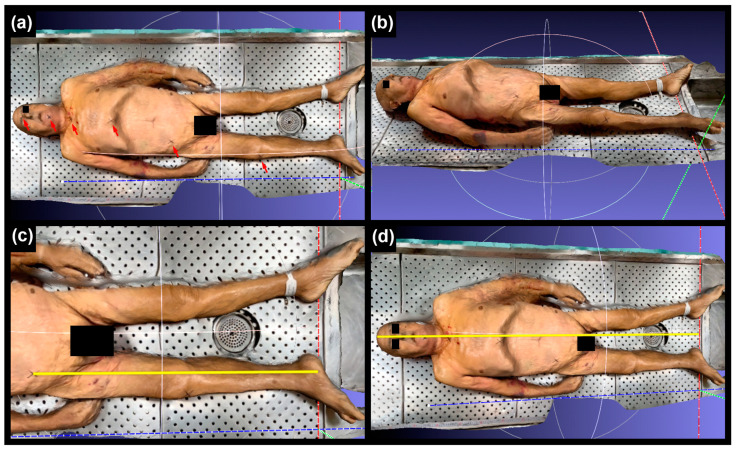
(**a**,**b**) Use of “draw XYZ axes in world coordinates” function in MeshLab (red arrows indicated the 6 anthropometric landmarks). (**c**,**d**) Use of the “reference scene tool” function in MeshLab to measure longitudinal distances.

**Figure 3 sensors-24-00500-f003:**
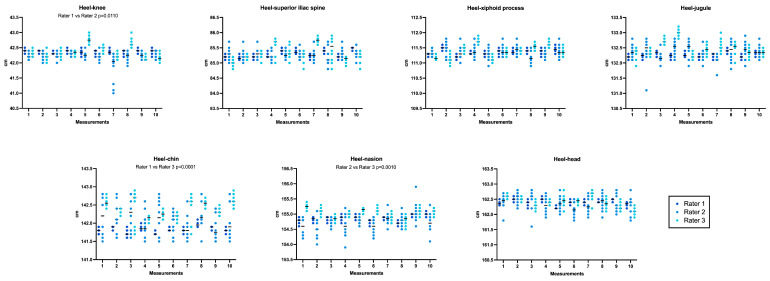
Graphical representation of the 7 digital measurements conducted by 3 indipendent raters.

**Figure 4 sensors-24-00500-f004:**
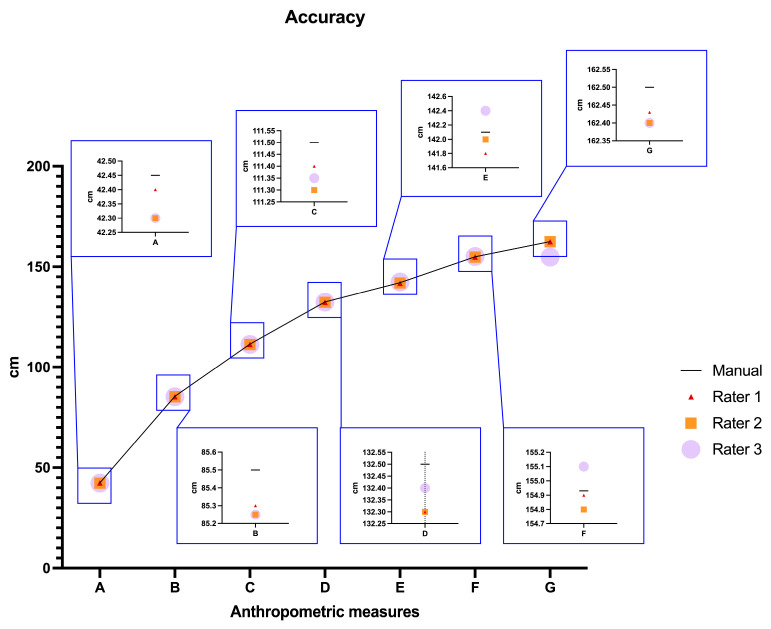
Accuracy assessed for each rater.

**Table 1 sensors-24-00500-t001:** Manual and digital anthropometric measurements, together with intra-rater reliability and intraclass correlation (ICC).

Anthropometric Measurement	Median Manual Measurement (IQ) (cm)	Digital Measurement				
		Intra-Rater Reliability			ICC	
		Median (IQ) (cm) Significant Differences	Median (IQ) (cm) Significant Differences	Median (IQ) (cm) Significant Differences	Single Measures	Average Measures
A. Heel–knee	42.45 (0.05)	42.40 (0.05) *p* = 0.5330	42.20 (0.07) *p* = 0.8822	42.30 (0.33) *p* = 0.4384	1.000	1.000
B. Heel–superior iliac spine	85.50 (0.05)	85.50 (0.20) *p* = 0.1022	85.20 (0.09) *p* = 0.8532	85.39 (0.44) *p* = 0.2025		
C. Heel–xiphoid process	111.50 (0.01)	111.38 (0.11) *p* = 0.6143	111.32 (0.11) *p* = 0.9565	111.38 (0.40) *p* = 0.2854		
D. Heel–jugule	132.50 (0.05)	132.26 (0.12) *p* = 0.6417	132.39 (0.16) *p* = 0.9320	132.44 (0.22) *p* = 0.0980		
E. Heel–chin	142.10 (0.10)	141.80 (0.13) *p* = 0.1393	142.10 (0.27) *p* = 0.1919	142.47 (0.36) ***p* = 0.0136** *		
F. Heel–nasion	154.93 (0.10)	154.90 (0.13) *p* = 0.1045	154.67 (0.31) *p* = 0.6108	155.10 (0.22) *p* = 0.2814		
G. Heel–head	162.50 (0.06)	162.4 (0.14) *p* = 0.8864	162.39 (0.15) *p* = 0.5401	162.41 (0.25) *p* = 0.5456		

* Bolded *p*-value is significant.

**Table 2 sensors-24-00500-t002:** An overview of accuracy for digital anthropometric measurements A to G, as calculated for each rater and for the median among them compared to the manual measurement.

Anthropometric Measurement	Accuracy Max Percent Error (Median)			
	Rater 1	Rater 2	Rater 3	Median among Raters
A. Heel–knee	<0.6% (0.1) Excellent	<0.8% (0.4) Excellent	<1.3% (0.4) Excellent	<0.6% (0.4) Excellent
B. Heel–superior iliac spine	<0.5% (0.2) Excellent	<0.6% (0.3) Excellent	<0.8% (0.3) Excellent	<0.4% (0.2) Excellent
C. Heel–xiphoid process	<0.4% (0.1) Excellent	<0.5% (0.2) Excellent	<0.5% (0.1) Excellent	<0.2% (0.1) Excellent
D. Heel–jugule	<0.3% (0.2) Excellent	<0.5% (0.2) Excellent	<0.5% (0.1) Excellent	<0.2% (0.2) Excellent
E. Heel–chin	<0.4% (0.2) Excellent	<0.5% (0.1) Excellent	<0.6% (0.2) Excellent	<0.1% (0.1) Excellent
F. Heel–nasion	<0.2% (0.0) Excellent	<0.5% (0.1) Excellent	<0.3% (0.1) Excellent	<0.1% (0.0) Excellent
G. Heel–head	<0.2% (0.0) Excellent	<0.5% (0.1) Excellent	<0.4% (0.1) Excellent	<0.1% (0.1) Excellent

## Data Availability

Data are contained within the article.

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
