# Peer review of "The Precision, Inter-Rater Reliability, and Accuracy of a Handheld Scanner Equipped with a Light Detection and Ranging Sensor in Measuring Parts of the Body—A Preliminary Validation Study"

_sensors, 2024, doi:10.3390/s24020500_

Round 1
Reviewer 1 Report
Comments and Suggestions for Authors
The structure of this article supports the stated primary objective: to present an alternative scanning method using a modern device, the iPadPro. It is essential to understand and know that body scans can be performed either with mobile devices or with stationary devices (3D scanning).
The authors emphasise the usefulness of a new device, the iPad Pro, for the scanning process in medico-legal practices.
I recommend to revise and clarify the following details:
-line 70..... is written: "The blind raters consisted of a forensic pathologist with 15 years of experience (GV), a forensic pathologist specializing in anthropology (PF), and a resident in Forensic Medicine (EC). "
The raters were blind????? How did they perform the needed activities?
-line 75... please explain "n total, 700 digital measurements (7 distances tested on 10 different scans, each measured ten times) were conducted".
I understood it to mean that each rater measured the same 7 distances over a period of 10 days... that is, 70 in total, multiplied by 3 equals about 210.
The text talks about 700 digital measurements. From where? And how?
The measurements were repeated? Why?
It is not recommended that the experience and practise of the experts have any influence on the accuracy of the results. The experts must have the same experience and understanding of the scanning protocol, regardless of whether they measure using the direct or scanning method.
-Line 102.... Explain the scanning procedure and details. Where was it placed, and how was it used? What were the differences between the known scanning protocol (for living persons) and this new solution?
-see measurement units in Table 1. I think that the unit is centimetre, not mm.
-line...195.....correct the word"infipendent"
-​it was perhaps better to compare the results and scanning process of the iPad-Pro with other mobile scanners available on the market. This iPad-Pro can be a useful solution, among others.
I recommend mentioning some technical details of this mobile solution, but compared to others already used and known in this field - the medico-legal field. What is the novelty or efficiency of this new mobile scanner?
-section Discussion.... it is said, " Currently, manual measurements using metersticks and callipers represent the “gold standard” in forensic investigations [7]" .Then it is said, "This was necessary because manual measurements conducted between two body landmarks can lack precision, as these landmarks might be situated at different levels of the body. The measurement performed matching the meter stick and the laser level exhibited statistically insignificant intra-rater variability." I see and understand a contradiction...
This section also states that there are various mobile solutions for body scanning. It was better to compare the results obtained with this scanner with those obtained with other scanning solutions.
Comments on the Quality of English Language
I have noticed some minor linguistic problems, which I have already mentioned in the previous section.
-line...195.....correct the word"infipendent"
Reviewer 2 Report
Comments and Suggestions for Authors
In this study, the authors examined the use of a portable 3D optical scanning system using a LiDAR camera (iPad) with the Polycam scanning software. They found that the agreement of scans performed by different assessors was strong and similar to those measures performed by trained medical anthropometrics.
Comments to the authors
-More clarity on the 3D scan app is needed to make it clear how scans are taken and processed. In line 108, it hints that photos are taken (how many, how is this instructed in the app, etc.) but then line 111 states that a scan was performed- was this using a video, multiple image captures, etc.? Does the app explain how the images (or video capture) are stitched to create the mesh? If possible, this would be helpful to add.
-Not sure why sections 2.3 and 2.4 are both body scan.
-The abstract states that the Polycam app provided accurate measures but the methods indicate that it was Meshlab that was used to process images and derive the anthropometric measures? This needs to be clarified throughout and then help readers identify whether this means they are reliant upon the Polycam app or if a mesh can be captured through other software and then analyzed using MeshLab.
-It is unclear what is being stated in 2.5 about “images were performed using Prism,” please clarify if this means the images for the figures or part of the analyses.
-Table 1 F. Heel-Nasion digital #1 seems to show a non-significant value but it states “yes” for significance. In addition to fixing this, it may be easier to present results by removing the yes/no and simply bolding the significant p value and reported in the footnote that bolded values are significant.
-Line 217 would benefit from a citation to show how these scanners are being used in clinics. It may be most helpful to clarify how both accuracy and precision are essential aspects of 3DO systems for clinical use and the advantage of this 3D approach used is that it may reduce both intra- and inter-rator errors. An example:
Bennett JP, Liu YE, Quon BK, Kelly NN, Wong MC, Kennedy SF, Chow DC, Garber AK, Weiss EJ, Heymsfield SB, Shepherd JA. Assessment of clinical measures of total and regional body composition from a commercial 3-dimensional optical body scanner. Clin Nutr. 2022;41(1):211-218.
-Lines 230-235 should be presented as a single paragraph and further clarify both the software used by Fineschi as well as if the Polycam or MeshLab apps were the tools being evaluated for their measurement capabilities.
-Line 249- are these larger errors in smaller landmarks due to absolute or proportional error?
-Line 256 should be the 2nd sentence of the following paragraph
-There should also be a paragraph in the discussion about the magnitude of errors observed between the system and the absolute measurements. Though small, it needs to be recognized that small differences do exist. Then, it should be stated whether these amounts of errors are considered potentially clinically significant (as opposed to statistically significant). For example, is there a risk for error in classification if a 0.5-1mm error exists? Or if not, please state that.
Comments on the Quality of English LanguageOverall high quality, however a few errors that could be improved with an external review.
Reviewer 3 Report
Comments and Suggestions for Authors
This article uses the iPad laser scanning function to measure the length of each part of the corpse, and designs an experiment to compare with the method in this article from two aspects: different measurement personnel and different measurement methods. Reading the paper, there are some puzzling points: 1) From the perspective of measurement methods, the method in this article is somewhat innovative, but in Section 2, the article seems to only record the use process of the iPad software, lacking a description or innovation of the method principle. 2) This article measures the lengths of several parts of the body, but neither the introduction nor the methods mention what the purpose of measuring these lengths is. 3) In Section 4, the article discusses a large number of existing research methods and lacks discussion of experimental results, which is inconsistent with the structure of general papers. 4) The overall writing method of the article is like an experimental report
Section 1: The introduction is too simplistic. The article uses the LiDAR sensor equipped with iPad to scan corpses. What aspects of data are measured, what results are obtained, and what problems need to be solved? What are the advantages of this method?
Section 2: Is there any corresponding mathematical formula for the representation of distance in the coordinate system and the accuracy evaluation index?
Section 2.3 and Section 2.4 have the same title
Section3: In the experimental part, the author only showed some longitudinal measurement data of the human body, and the analysis of the experimental results did not reflect the characteristics or advantages of the method in this article.
Section 4: The discussion part is a review based on the properties shown by the experimental results, but the author introduces most of the existing research, which does not conform to the structure of a general paper
Round 2
Reviewer 1 Report
Comments and Suggestions for Authors
The authors have taken the recommendations into account.
Author Response
Thanks
Giovanni Cecchetto
Reviewer 3 Report
Comments and Suggestions for Authors
This revised manuscript adds some content to the original text, but there still remains some major problems.
Section 1:
The revisions in the introduction section are insufficient, and the authors still have not made it clear what problem this paper needs to address and what the unique strengths of the methodology are.In addition, there are errors or unrepresentative citations, for example, document 11, published in 1999, is about the rules of forensic autopsy, and I do not agree with the author's statement that the document, "A recent study [11] tested an Apple® tablet equipped with a LiDAR sensor, generating three-dimensional (3D) reconstructions of ten bodies directly within the autopsy room." In particular, there are also some documents that have such a mismatch with the meaning expressed in the article, so please be aware of this.
Section result:
In this part, the authors briefly discuss the experimental results, according to the experimental results there is still a certain gap between its accuracy and the manual measurement, for example, in “Heel-knee “, the maximum difference between the laser measurement and the manual measurement result is 0.25cm, which is a big error for the human body structure, so the advantage of this method is not reflected. In addition, there were also errors between the raters, but the authors did not explain the reasons for this phenomenon.
Section 4:
This section is still very confusing and is more like an introductory section. The authors present most of the existing studies rather than an overview based on the properties shown by the experimental results, and it is recommended that the authors refer to the way other papers are written.
Line221-274: This section discusses the research significance, existing methods and the advantages of LiDAR, which should appear in the introduction of Section 1
Line279-313: This section discusses the superficial results of the experiment, which should appear in the experimental of Section 3
Line326-329: As the authors note, the accuracy of the measurements relies on the “senior pathologist”. In other words, despite using an iPad, the measurement results are still affected by people's subjective consciousness. So what are the advantages of this method in measuring length compared with the "gold standard" (manual measurement), the article does not reflect it. The manual measurement method seems to be more efficient. Does the author consider reducing this manual intervention and achieving automatic measurement? These should appear in the Discussion section of Section 4
Author Response
Please see the attachment.
Thanks
Giovanni Cecchetto
